# Dispensing of antibiotics without prescription and associated factors in drug retail outlets of Eritrea: A simulated client method

**Merhawi Bahta**[1], **Sirak Tesfamariam**[2], **Dawit G. Weldemariam**[3]*, **Hermella Yemane**[4], **Eyasu H. Tesfamariam**[5], **Tesfamariam Alem**[1], **Mulugeta Russom**[6]

**1** Department of Medical Sciences, Pharmacy Unit, Orotta College of Medicine and Health Sciences, Asmara, Eritrea, **2** Pharmacy, Nakfa Hospital, Nakfa, Eritrea, **3** Pharmacy, Hazhaz Zonal Referral Hospital, Asmara, Eritrea, **4** Southern Red Sea Zonal Medical Store, Assab, Eritrea, **5** Department of Statistics, Biostatistics and Epidemiology Unit, College of Science, Eritrean Institute of Technology, Mai Nefhi, Eritrea, **6** Eritrean Pharmacovigilance Center, National Medicines and Food Administration, Asmara, Eritrea

* dawitghiwet100@gmail.com

**Data Availability Statement:** All relevant data are within the manuscript and its Supporting Information files.

## Abstract

Dispensing antibiotics without prescription is irrational and can hasten the emergence and spread of antibiotic resistance. This study aims at determining the extent of this practice and its determinants in all drug retail outlets of Eritrea. A cross-sectional simulated client method was used to conduct the study. Data was collected between July and August 2019, entered and analyzed using Statistical Package for Social Science version 22. Descriptive analysis was performed using mean (standard deviation), median (interquartile range), frequency, percentage, as appropriate, for independent variables. Logistic regression, at bivariate and multivariate levels, along with odds ratio (95% confidence interval) was used to determine the association between the dispensing of antibiotics without prescription and independent variables. *P*-values less than 0.05 were considered as statistically significant. The extent of dispensing antibiotics without prescription was found to be 87.6% with the most frequently dispensed antibiotics being ciprofloxacin (47.8%) and co-trimoxazole (37.5%). Furthermore, 12.4% of the drug retail outlet attendants did not dispense antibiotics because they preferred a referral to health facilities (52.6%), were following administrative restrictions not to sell antibiotics (42.1%), or did not have the necessary antibiotics (31.6%). Private community pharmacies (AOR = 7.68, 95% CI: 1.67, 35.37; *p* = 0.009) and private drug shops (AOR = 10.65, 95% CI: 1.96, 57.93; *p* = 0.006) were more likely to dispense antibiotics compared to the governmental community pharmacies. Dispensing antibiotics without prescription was more likely to occur in the Maekel (central) region (AOR = 3.76, 95% CI: 1.19, 11.92; *p* = 0.024) compared to the remaining regions combined. In conclusion, the sales of antibiotics without prescription in the drug retail outlets of Eritrea is alarming which requires immediate attention from policymakers.

**Funding:** The authors received no specific funding for this work.

**Competing interests:** The authors have declared that no competing interests exist.

# Introduction

Antibiotic resistance is a global public health threat that warrants immediate global intervention [1, 2]. To tackle this problem, the WHO, in its 68[th] world health assembly, passed a resolution, based on the global action plan, that instructs all member states to develop a national strategic plan on combating antimicrobial resistance [3]. One of the strategic objectives of the global action plan is to optimize the rational use of antimicrobials.

In line with the WHO resolution, Eritrea developed a national action plan to combat antimicrobial resistance. Moreover, the National Medicines and Food Administration in collaboration with stakeholders developed medicines schedule guide in June 2019. The medicines are scheduled by the need for prescription and their availability in drug retail outlets. On top of the above initiatives, Eritrea has been conducting annual campaigns on awareness of antibiotic resistance using a one-health approach since 2017.

Self-medication and antibiotics dispensing without prescription in drug retail outlets [4–7] are irrational practices that can accelerate the emergence and spread of antibiotic resistance [8]. It has been estimated that more than 50% of antibiotics worldwide are sold without prescription [9]. Although dispensing antibiotics without prescription in the developed world is minimal [10–14], the problem is much higher in Africa and other low-income countries due to poor healthcare regulations [12]. In Eritrea, to the best of the authors' knowledge, the prevalence of dispensing antibiotics without prescription is not known. However, a recent study conducted in Asmara, the capital city of Eritrea, reported a significant rate (45.1%) of self-medication with antibiotics [15].

In most countries, as antibiotics are accessed mostly through community pharmacies, intervening at the practice of these drug retail outlets can minimize inappropriate use of antibiotics including self-medication and sales without prescription [16–18]. This study, therefore, was conducted to determine the extent of dispensing antibiotics without prescription and its associated factors in Eritrea. The results will be helpful to programmers, policymakers, regulators and other concerned bodies to know the burden of the problem and identify areas of intervention.

# Materials and methods

## Study design and setting

A cross-sectional investigational study using simulated client method was conducted between July and August 2019 in all but four drug retail outlets available in Eritrea. Totally, 75 private and 13 governmental drug retail outlets (49 pharmacies and 39 drug shops) are located in all six administrative regions of Eritrea. The four drug retail outlets were excluded either due to budget constraints, as two of them were located in hard-to-reach areas, or conflict of interest.

## Study population

In Eritrea, the professional in charge of a pharmacy is a pharmacist, while those in charge of drug shops are either a pharmacist or pharmacy technician. This study included 84 pharmacies and drug shops which were open to the public during the study period. The dispensers of these drug retail outlets during the visit were considered as the study population.

## Data collection tool and approach

To capture the routine performance of dispensing, dispensers were blinded using simulated clients to collect data. Simulated client method is an effective method of deriving valid

measures of healthcare providers' actual practices, which are challenging to achieve through any other method [19, 20].

Data was collected by six simulated clients (three female and three male final year pharmacy students), not identifiable by the dispensers. To ensure uniform presentation and consistency on their level of demand (data collection), the simulated clients were well-trained and rehearsed the role plays for two days. Prior to the data collection, the simulated clients were asked to fill a non-disclosure confidentiality form to conduct the study in strict confidence and avoid disclosure of any malpractice of the dispensers to other parties.

The clients simulated the same case with the same signs and symptoms and provided consistent answers for queries raised by the dispensers. Female simulated clients were acting as patients with a case of uncomplicated urinary tract infection (UTI) presenting with a chief complaint of "burning sensation upon urination for 3 consecutive days." If further asked by the dispenser, other aspects of the medical history included the presence of dysuria, frequency of urination, urgency, supra-pubic pain, and sensation of bladder fullness or lower abdominal discomfort with the absence of fever, chills, vaginal discharge, flank pain, allergies, and pregnancy. On the other hand, the male simulated clients acted as patients with acute watery diarrhea experiencing "loose bowel motion accompanied by diarrhea for 3 days". Further medical history on request included the presence of watery stool, loss of appetite, generalized body weakness, slightly increased bowel sounds, and a four times frequency of bowel movement per day with absence of blood/pus in the stool, vomiting, fever, abdominal tenderness, and allergies. These two clinical cases were selected since they are common ailments in the general population, antibiotics may not be warranted in all patients with those symptoms, and if needed, laboratory aided diagnosis is required for confirmation of the etiologic pathogen before antibiotic prescription.

Each drug retail outlet was visited twice (one for each case) with a reasonable time gap between the visits in order to avoid suspicion from dispensers. The simulated clients were accompanied by another colleague to minimize recall bias and improve the reliability of the information provided during the encounter. Three gradually incremental levels of demand were used to convince the drug retail outlet attendants to get an antibiotic without prescription. The details of the visit process are depicted in Fig 1.

The data recording form (S1 File) comprises three sections including questions on demographic information of both the drug retail outlets and dispensers, the response of the dispenser to the request of the simulated clients, and comments, such as the appearance of the dispenser to help with the identification, that couldn't be filled in the other sections. Two days prior to the actual study, a pre-test for the questionnaire was conducted in four drug retail outlets of Asmara. The data gathered from these sites were included in the analysis since no change was made on the data recording form and data collection approach.

For each visited pharmacy, the database of the Licensing office, National Medicines and Food Administration was used to retrieve information that includes the exact age, profession and work experience of a dispenser.

## Ethical consideration

Ethical clearance to conduct the study was obtained from the National Medicines and Food Administration of the Ministry of Health, the authority to grant permission for such investigational studies. All ethical and professional considerations were followed throughout the study to keep the data strictly confidential. The names of the drug retail outlets and identifiers of the dispenser are kept confidential. Investigational information will only be disclosed at the aggregate level and all findings at a personal level will be kept confidential.

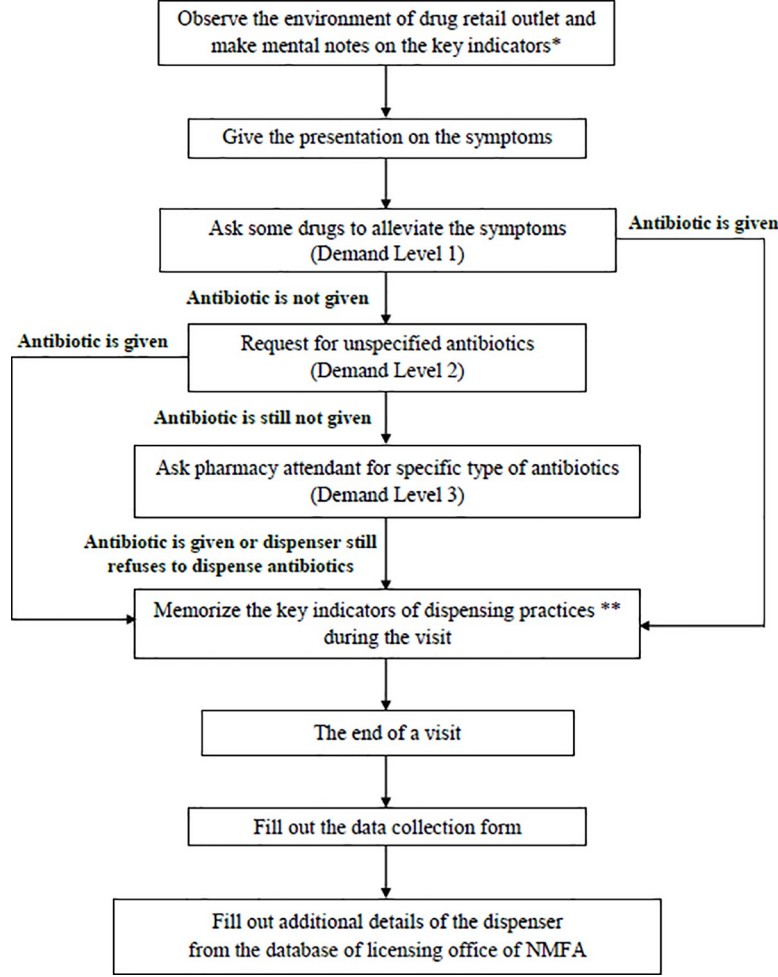

**Fig 1. Data collection process.** *key indicators*\* included name tag, professional license, the appearance of the dispenser *key indicators of dispensing practices* \*\* included the name of antibiotics dispensed (if any), demand level, reasons for not dispensing, and alternative medications offered (if any) *NMFA*: National Medicines and Food Administration.

## Statistical analysis

Data was entered and analyzed using the Statistical Package for Social Science version 22 (SPSS-22). Both descriptive and analytical analysis were carried out. Descriptive analysis was performed using mean (SD) or median (IQR) as appropriate for continuous variables. Categorical variables were summarized using frequency and percentage. The association between the dispensing of antibiotics without prescription and variables (type and location of the drug retail outlet, age, sex, work experience and profession of the dispenser, and type of clinical scenario) was explored using bivariate and multivariate logistic regression. Variables that were found to be significant at bivariate level were included at the multivariate level. Odds ratio with 95% confidence interval was calculated for both bivariate and multivariate associations. *P*-values less than 0.05 were considered as significant throughout the study.

## Results

A total of 153, out of the expected 168 questionnaires, were fully completed with a response rate of 91.1%. The majority of dispensers studied (64.1%) were males. The study population

**Table 1. Socio-demographic characteristics of the dispensers.**

| Variables | | Frequency | Percent |
|---|---|---|---|
| **Sex** | | | |
| | Male | 98 | 64.1 |
| | Female | 55 | 35.9 |
| **Title of the dispenser** | | | |
| | Pharmacist | 55 | 35.9 |
| | Pharmacy technician | 64 | 41.8 |
| | Nursing degree | 2 | 1.3 |
| | Nursing diploma | 5 | 3.3 |
| | Health assistant | 17 | 11.1 |
| | Nurse aid | 1 | 0.7 |
| | Non professional | 9 | 5.9 |
| **Age** | (Mean = 48.6, SD = 14, Range = 24 to 83) | | |
| **Work experience** | (Median = 10, IQR = 14, Range = 0 to 47) | | |

comprised 41.8% pharmacy technicians, 35.9% pharmacists and others (Table 1). The mean age of the dispensers was found to be 48.6 years (SD = 14) with a median dispensing experience of 10 years (IQR = 14).

The extent of dispensing antibiotics without prescription was found to be 87.6%. Specifically, for uncomplicated UTI and acute watery diarrhea, antibiotics were dispensed in 89.2% and 86.1% of the cases respectively. The frequently dispensed antibiotics were ciprofloxacin (47.8%) and co-trimoxazole (37.5%) (Table 2). For individual cases, co-trimoxazole (53.6%) in acute watery diarrhea and ciprofloxacin (56.7%) in uncomplicated UTI were the most dispensed. Moreover, 81.3% of the dispensed antibiotics were given at demand level one, 11.2% at demand level two, and 6.7% at demand level three.

In 12.4% of the simulated encounters, antibiotics were not dispensed without prescription. In 52.6% of these encounters, the dispenser chose to refer the patient for further clinical investigation instead of dispensing antibiotics without prescription. Other reasons for not dispensing were administrative restrictions (rules and regulations set by individual drug retail outlets for their own practices) not to sell antibiotics without prescriptions (42.1%) and unavailability of antibiotics during the visit (31.6%). However, other than antibiotics, loperamide and oral rehydration salts were provided to a few clients.

Chi-square test was done to see if there is any association between antibiotics dispensed without prescription and the independent categorical variables namely sex and profession of the dispenser, type of the drug retail outlet (private community pharmacy, private drug shop,

**Table 2. Percentage of the antibiotics dispensed without prescription.**

| Antibiotics dispensed | Acute Watery Diarrhea | | Acute Uncomplicated UTI | | Total | |
|---|---|---|---|---|---|---|
| | Frequency | Percent | Frequency | Percent | Frequency | Percent |
| Ciprofloxacin | 27 | 39.1 | 38 | 56.7 | 65 | 47.8 |
| Co-trimoxazole | 37 | 53.6 | 14 | 20.9 | 51 | 37.5 |
| Amoxicillin | 0 | 0 | 11 | 16.4 | 11 | 8.1 |
| Doxycycline | 2 | 2.9 | 3 | 4.5 | 5 | 3.7 |
| Tinidazole | 2 | 2.9 | 1 | 1.5 | 3 | 2.2 |
| Metronidazole | 1 | 1.4 | 0 | 0 | 1 | 0.7 |
| **Total** | 69 | 100 | 67 | 100 | 136 | 100 |

or governmental community pharmacy), regional location of the drug retail outlet, and type of clinical scenario. Among the tested variables; regional location of the drug retail outlet ($p = 0.008$) and type of the drug retail outlet ($p < 0.0001$) were found to be significantly associated with dispensing antibiotics without prescription.

A binary logistic regression was performed to observe the magnitude of the influence and to test continuous variables (age and work experience). In the bivariate analysis, type of the drug retail outlet ($p < 0.0001$), regional location ($p = 0.012$), and age of the dispenser ($p = 0.019$) were found to significantly influence the dispensing of antibiotics without prescription. Even though the association with work experience was found to be insignificant ($p = 0.185$), the factor of experience was included in the multivariate level as it might have an effect on the other variables (Table 3).

To adjust for confounding variables, multivariate analysis was performed for variables that were found to be significant at bivariate level. The result shows that the type of drug outlet and regional location retained their significance. It was found that private community pharmacies (AOR = 7.68, 95% CI: 1.67, 35.37) and private drug shops (AOR = 10.65, 95% CI: 1.96, 57.93) were more likely to dispense antibiotics without prescription compared to the governmental community pharmacies. Regarding the location of the drug retail outlets, those located in Maekel (central) region, which include facilities in the capital city, had a higher inclination of dispensing antibiotics without prescription (AOR = 3.76, 95%CI: 1.19, 11.92) compared to the remaining outlets in the other five regions combined (Table 3).

## Discussion

Dispensing antibiotics without prescription in drug retail outlets available in Eritrea was found to be commonly practiced. The result of this study reveals easy access to antibiotics which can, in turn, accelerate the development of antibiotics resistance [21]. Furthermore, this can result in wastage of resources, wrong choice of antibiotics, and experience of adverse drug reactions. Dispensing of antibiotics without prescription was found to be less frequent in studies conducted in Gipuzkoa, a Spanish province (17.5%) [11], Republic of Srpska (18.5%) [22], northwestern Spain (18.83%) [23], Beirut and its suburbs, Lebanon (32%) [24], Sri Lanka (41%) [25], Catalonia, Spain (45.2%) [26], and urban cities of China (66.8%) [10]. A result extracted from a study done in Albania (80%) [14] was comparatively consistent with findings of the current study, but higher results were observed in studies done in Moshi municipality, Tanzania (92.3%) [12], Mizan-Aman town, southwest Ethiopia (94.4%) [27], and in Lusaka district, Zambia (100%) [13]. The discrepancy in the magnitude observed might be due to differences in the simulated cases, the methodological approach used, and sample size. These findings

**Table 3. Bivariate and multivariate analysis on the association of independent variables and dispensing antibiotic without prescription.**

| Variables | Bivariate analysis | | | Multivariate analysis | | |
|---|---|---|---|---|---|---|
| | COR | 95% CI | *p*-value | AOR | 95% CI | *p*-value |
| **Type of drug outlet** | | | | | | |
| Private community pharmacy (n = 62) | 11.15 | (3.02, 41.18) | <0.0001 | 7.68 | (1.67, 35.37) | 0.009 |
| Private drug shop (n = 68) | 9.69 | (2.84, 33.11) | <0.0001 | 10.65 | (1.96, 57.93) | 0.006 |
| Governmental community pharmacy (n = 23) | *Ref* | | | *Ref* | | |
| **Location of drug outlet** | | | | | | |
| Maekel region (n = 98) | 3.63 | (1.33, 9.86) | 0.012 | 3.76 | (1.19, 11.92) | 0.024 |
| Other regions (n = 55) | *Ref* | | | *Ref* | | |
| **Age of the dispenser** | 1.05 | (1.01, 1.09) | 0.019 | 1.000 | (0.94, 1.06) | 0.933 |
| **Year of experience** | 1.04 | (0.98, 1.09) | 0.185 | 1.025 | (0.96, 1.10) | 0.494 |

together with our study's result; however, reveals frequent misuse of antibiotics in drug retail outlets across broad regions of the world, especially in Africa.

Even though the root causes may be accurately identified by a qualitative study, weak regulation of antibiotic sales, lax enforcement of existing policies on antibiotics dispensary, and unavailability of medicines scheduling could be the main causes of pervasive non-prescription access to antibiotics. Most of the antibiotics were dispensed at demand level one, which reflects the current study's dispensers' leniency towards the sales of medicines in general, and antibiotics in particular. A similar case was observed in a study done in Ethiopia, where most antibiotics were dispensed when the simulator asked for any medication to alleviate his/her symptoms [27]. Inadequate training targeted towards authorized dispensers about rational use of antibiotics and high patient demand of seeking medications to avoid lengthy time in health facilities due to over-stretched healthcare systems in developing countries [28] may have contributed to the willingness of the dispensers to give antibiotics. Continuing professional development and on-job training of drug dispensers should be routinely emphasized to mitigate these problems.

In the present study, governmental community pharmacies were found to be less inclined to dispense antibiotics and administrative restriction was the main reason reported for not dispensing antibiotics without prescription. Additionally, several business licenses of drug retail outlets in Eritrea are owned by non-professionals, thus owners might put pressure on the licensed pharmacy professionals for the sake of economic benefits and this may compromise the decision of the professionals.

The difference in dispensing antibiotics without prescription was influenced by the location of the drug retail outlets, which was higher in Maekel region compared to the other regions combined. Unavailability of antibiotics during the study period was the main reason that dispensers in the other region claimed for not dispensing antibiotics. The *"other regions combined"* have a wide difference in their location, and hence the difference in the dispensing behavior could be due to unexplored factors.

No difference in the dispensing antibiotics without prescription was observed according to the profession of the dispensers. Even though the distribution shows high involvement of unauthorized dispensers including non-healthcare professionals, the authorized dispensers; pharmacists and pharmacy technicians, did not appear to impact appropriate practice. Ciprofloxacin was the most frequently dispensed antibiotic, especially in uncomplicated UTI similar to findings from an Ethiopian [27] and a Sri Lankan [25] studies. Ciprofloxacin is in the watch group of WHO antibiotics category [21] and being from the fluoroquinolone class, cross-resistance to other fluoroquinolone drugs is challenging, especially for those used in the second-line anti-TB treatment [29]. This shows that pharmacists' role in combating antibiotic resistance was not appreciable, and warrants continuous education, sensitization, and strengthening the regulation on the rational use of antibiotics.

## Limitation of the study

The case scenarios selected for the study could have influenced the extent of the dispensing antibiotics practice. Simulating other cases like upper respiratory tract infections (acute otitis media, rhinosinusitis, acute pharyngitis), lower respiratory tract infections, or tonsillitis might have given a different picture of the practice.

## Conclusion and recommendations

Sales of antibiotics without prescription in Eritrean drug retail outlets is alarming which can escalate the issue of antibiotic resistance and compromise the availability of antibiotics in the country. Strengthening existing laws, guidelines and national medicines regulatory capacity,

continuous awareness-raising programs, and implementation of medicines scheduling are recommended to achieve more prudent use of antibiotics. Besides, future regulatory interventions should consider the access and excess use of antibiotics. Further qualitative study is also suggested to unveil reasons that motivate the sale of antibiotics without prescription.

## Supporting information

**S1 File. Data recording form.**
(PDF)

## Acknowledgments

The authors would like to thank the data collectors, NMFA as well as ERIPA's support in conducting the study. The authors' sincere gratitude also goes to Mr. Meswaiti Teklezghi, Mr. Tesfasilasie Ghedel, and Mr. Michael Ghirmay for their technical assistance in retrieving the detailed information of the dispensers.

## Author Contributions

**Conceptualization:** Mulugeta Russom.

**Data curation:** Merhawi Bahta, Sirak Tesfamariam, Eyasu H. Tesfamariam.

**Formal analysis:** Merhawi Bahta, Sirak Tesfamariam, Hermella Yemane, Eyasu H. Tesfamariam, Tesfamariam Alem, Mulugeta Russom.

**Investigation:** Merhawi Bahta, Dawit G. Weldemariam, Mulugeta Russom.

**Methodology:** Merhawi Bahta, Sirak Tesfamariam, Dawit G. Weldemariam, Hermella Yemane, Eyasu H. Tesfamariam, Tesfamariam Alem, Mulugeta Russom.

**Project administration:** Merhawi Bahta, Mulugeta Russom.

**Resources:** Mulugeta Russom.

**Supervision:** Merhawi Bahta, Sirak Tesfamariam, Dawit G. Weldemariam, Tesfamariam Alem, Mulugeta Russom.

**Validation:** Mulugeta Russom.

**Writing – original draft:** Merhawi Bahta, Sirak Tesfamariam, Dawit G. Weldemariam, Hermella Yemane, Tesfamariam Alem.

**Writing – review & editing:** Merhawi Bahta, Sirak Tesfamariam, Dawit G. Weldemariam, Hermella Yemane, Eyasu H. Tesfamariam, Tesfamariam Alem, Mulugeta Russom.

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
