## [Decision Letter · Decision Letter 0]

20 Nov 2019

PONE-D-19-28324

Prevalence of antibiotic dispensing without prescription and associated factors in drug retail outlets of Eritrea: a simulated client method

PLOS ONE

Dear Dr. Weldemariam

Thank you for submitting your manuscript to PLOS ONE. After careful consideration, we feel that it has merit but does not fully meet PLOS ONE’s publication criteria as it currently stands. Therefore, we invite you to submit a revised version of the manuscript that addresses the points raised during the review process.

Please address each of the comments raised below by the reviewer.  In addition, I would ask you to respond to the following concerns:

1. Line 93 - it appears that you surveyed ALL pharmacies and drug shops in Eritrea.  Please clarify

2. Lines 195 - 200 - please expand this.  Zonal location and type of pharmacy were associated with dispensing, but it is not clear what that means.  What is a zonal location?  Which ones had which effect?  What kind of pharmacy was associated with dispensing?

3. Lines 223-238 - This entire paragraph is confusing.  It seems to say that the likelihood of dispensing without a prescription is lower in some areas and higher in others.  Referring to this as prevalence is confusing.

4.  The paper needs a Limitations section.  What should the reader be aware of when determining the validity and accuracy of the paper?

5. References - Reference 13 has no journal listed.  References 16 and 20 have extraneous information in the citation.

6.  You correctly point out that uncontrolled access to antibiotics has a number of significant concerns including increases bacterial resistance.  But it is also important to consider that strictly enforcing restrictions on drug sales in many parts of the Global South can have the unintended consequence of preventing patients who cannot afford a clinic or doctor from receiving any treatment at all.  What policy recommendations do you make as a result of your study?

We would appreciate receiving your revised manuscript by December 6, 2019. To enhance the reproducibility of your results, we recommend that if applicable you deposit your laboratory protocols in protocols.io, where a protocol can be assigned its own identifier (DOI) such that it can be cited independently in the future. For instructions see: http://journals.plos.org/plosone/s/submission-guidelines#loc-laboratory-protocols

We look forward to receiving your revised manuscript.

Kind regards,

John Rovers, PharmD, MIPH

Academic Editor

PLOS ONE

Journal Requirements:

2. Please ensure you have thoroughly discussed any potential limitations of this study within the Discussion section.

3. During your revisions, please note that a simple title correction is required to "Prevalence of antibiotic dispensing without prescription and associated factors in drug retail outlets of Eritrea: a simulated client method" due to a missing space in the current title. Please ensure this is updated in the manuscript file and the online submission information.

Reviewers' comments:

Reviewer's Responses to Questions

**Comments to the Author**

1. Is the manuscript technically sound, and do the data support the conclusions?

Reviewer #1: Yes

2. Has the statistical analysis been performed appropriately and rigorously? 

Reviewer #1: Yes

3. Have the authors made all data underlying the findings in their manuscript fully available?

Reviewer #1: Yes

4. Is the manuscript presented in an intelligible fashion and written in standard English?

Reviewer #1: Yes

5. Review Comments to the Author

Reviewer #1: Dear Authors,

I very much enjoyed reading your paper. Thank you for allowing me to review it. I have the following comments:

1) Around line 122- I think it’s important that you note that both of these disease states likely require laboratory aided diagnosis to confirm if it is a bacterial infection; make that connection so that the reader (especially if they are not a healthcare professional) understands that antibiotics may not be warranted in all patients with these symptoms. This is key as otherwise the reader may not connect the irrational use piece and would need to understand why the patient would need to go to the doctor for a workup vs just receiving the antibiotic.

2) Figure 1 - will the lines 131-136 go with the figure? Otherwise it is confusing.

3) Line 142: I believe you mean it was a “pilot test” of the data collection process/questionnaire vs a “pre-test” which would imply that there is a post-test

4) Line 192 and again lower, I’m confused about the term “administrative restrictions” operationalize this term in the paper

5) I would like to see the sample size for each of the types of pharmacies and pharmacy locations. Consider adding n= to each on Table 3

6) Line 224 – your study doesn’t prove if there is “excessive consumption” as your student doesn’t specifically measure if those with the symptoms may have an infection. I would add the word “potential” excessive consumption – or something similar

7) Line 239 – this is an assumption, change “can” to “may” or “might”

8) Line 269 – while you mention Cipro is on the WHO watch group, I think it is important to address the potential issues with TB and prescribing a drug from this class without consideration of TB.

9) As PLOS does not have a copyeditor, I would have someone who is skilled and is not one of the authors (for fresh eyes), give your paper a final proofreading. There are some issues with spacing - sometimes too many, sometimes not enough. On occasion there are dropped articles ("a"); that said, the articles could be a matter of British vs. American English. There are also some capitalization inconsistencies in table 1, and in your paper pharmacy should always be lowercase unless part of the name of a location. Eliminate the conjunction in line 140.

Overall, this is a well done study and well written paper.

6. PLOS authors have the option to publish the peer review history of their article (what does this mean?). If published, this will include your full peer review and any attached files.

Reviewer #1: No

---

## [Author Response · Author response to Decision Letter 0]

18 Dec 2019

Response to reviewers

We would like to thank the academic editor and reviewer for their thorough review and constructive comments. After carefully considering the points raised, we have made the following changes to the manuscript and put forward explanations as required.

Academic editor’s comments

1. Line 93 - it appears that you surveyed ALL pharmacies and drug shops in Eritrea. Please clarify

Response: Comment accepted. 

The survey was done in 84 pharmacies and drug shops (out of total 88), in which the four were excluded for several reasons. Accordingly, modification was made and can be referred in lines 88-90, 94-95.

2. Lines 195 - 200 - please expand this. Zonal location and type of pharmacy were associated with dispensing, but it is not clear what that means. What is a zonal location? Which ones had which effect? What kind of pharmacy was associated with dispensing?

Response: Accepted.

Eritrea is divided administratively into six zones (regions). And, zonal (regional) location mentioned in the manuscript stands for this administrative classification.As per the study’s result, the place where a drug retail outlet is located and a drug retail outlet being a private pharmacy, private drug shop, or governmental pharmacy were found to affect the dispensing behavior.The magnitude of the influence is further explained in lines 207-213.

3. Lines 223-238 - This entire paragraph is confusing. It seems to say that the likelihood of dispensing without a prescription is lower in some areas and higher in others. Referring to this as prevalence is confusing.

Response: Comment well taken.

We have edited the paragraph to make it more pellucid.

4. The paper needs a Limitations section. What should the reader be aware of when determining the validity and accuracy of the paper?

Response: Accepted

We have added one limitation as a sub section of the discussion. 

5. References - Reference 13 has no journal listed. References 16 and 20 have extraneous information in the citation.

Response: Accepted

We have removed the extraneous information from references 16 and 20. But for reference 13, “Expert review of anti-infective therapy” is the name of the journal cited.

6. You correctly point out that uncontrolled access to antibiotics has a number of significant concerns including increases bacterial resistance. But it is also important to consider that strictly enforcing restrictions on drug sales in many parts of the Global South can have the unintended consequence of preventing patients who cannot afford a clinic or doctor from receiving any treatment at all. What policy recommendations do you make as a result of your study?

Response: Accepted

We understand the concern that could emerge by enforcing strict restrictions on drug retails. But, the case is different in Eritrea. The fee for health services is nominal, and it is also free for those who claim who cannot afford at all. Moreover, sometimes it is also more expensive for services provided in drug retail outlets than in health facilities. Considering this fact in our country, we don’t believe the recommended restrictions could have considerable effect on preventing indigent patients from receiving treatments. We have also recommended a further research to unveil the reasons that motivate the sales of antibiotics and patients seeking for antibiotics from drug retails even when they can have an access to health facilities. 

Reviewer’s comments

1.Around line 122- I think it’s important that you note that both of these disease states likely require laboratory aided diagnosis to confirm if it is a bacterial infection; make that connection so that the reader (especially if they are not a healthcare professional) understands that antibiotics may not be warranted in all patients with these symptoms. This is key as otherwise the reader may not connect the irrational use piece and would need to understand why the patient would need to go to the doctor for a workup vs just receiving the antibiotic.

Response: Accepted

Thank you for the interesting comment. To make it clear, we have added in the paragraph that both the disease states require laboratory aided diagnosis to confirm the etiologic agent.

2. Figure 1 - will the lines 131-136 go with the figure? Otherwise it is confusing.

Response: Accepted

Yes, they go with the figure.

The title of the figure is “Figure 1: Data collection process. ”And, the legend of the figure is“Figure 1 Legend: key indicators* included name tag, professional license, appearance of the dispenser

key indicators of dispensing practices ** included name of antibiotics dispensed (if any),demand level, reasons for not dispensing, and alternative medications offered (if any) NMFA: National Medicines and Food Administration.”

3. Line 142: I believe you mean it was a “pilot test” of the data collection process/questionnaire vs a “pre-test” which would imply that there is a post-test

Response: Accepted

A pre-test study for the questionnaire was done to check the comprehensibility the data collection tool. As we conducted it in only 4 drug retail outlets, we cannot dare to call a pilot study. Since no modifications were made, it was not necessary to conduct a post-test. But, we have edited the sentence to indicate that the pre-test was done for the questionnaire, not for the data collection process.

4) Line 192 and again lower, I’m confused about the term “administrative restrictions” operationalize this term in the paper

Response: Accepted

As we have added in the paragraph, the term “administrative restrictions”indicates for the rules and regulations set by individual drug retail outlets for their own practices.

5) I would like to see the sample size for each of the types of pharmacies and pharmacy locations. Consider adding n= to each on Table 3

Response: Accepted

We have added the number of samples in the specified table 3.

6) Line 224 – your study doesn’t prove if there is “excessive consumption” as your student doesn’t specifically measure if those with the symptoms may have an infection. I would add the word “potential” excessive consumption – or something similar

Response: Accepted

We have eliminated the term “excessive consumption” from the sentence.

7) Line 239 – this is an assumption, change “can” to “may” or “might”

Response: Modification is done accordingly.

8) Line 269 – while you mention Cipro is on the WHO watch group, I think it is important to address the potential issues with TB and prescribing a drug from this class without consideration of TB.

Response: The comment is addressed well. (Refer lines 264-267)

9) As PLOS does not have a copyeditor, I would have someone who is skilled and is not one of the authors (for fresh eyes), give your paper a final proofreading. There are some issues with spacing - sometimes too many, sometimes not enough. On occasion there are dropped articles ("a"); that said, the articles could be a matter of British vs. American English. There are also some capitalization inconsistencies in table 1, and in your paper pharmacy should always be lowercase unless part of the name of a location. Eliminate the conjunction in line 140.

Overall, this is a well done study and well written paper.

Response: Thank you for the comments. The manuscript has been proofread by a skilled person and believed to be improved.

We believe that the comments are addressed and communicated thoroughly. Additionally, we would like to inform you that the line numbers given can only match with the cleaned revised manuscript. 

We are looking forward to receiving your decision at your earliest convenience. 

Kind regards,

Authors: Merhawi Bahta, Sirak Tesfamariam, Dawit G. Weldemariam, Hermella Yemane, Eyasu H. Tesfamariam, Tesfamariam Alem, and Mulugeta Russom

---

## [Editor Report · Decision Letter 1]

7 Jan 2020

Dispensing of antibiotics without prescription and associated factors in drug retail outlets of Eritrea: a simulated client method

PONE-D-19-28324R1

Dear Dr. Dawit G. Weldemariam

We are pleased to inform you that your manuscript has been judged scientifically suitable for publication and will be formally accepted for publication once it complies with all outstanding technical requirements.

With kind regards,

John Rovers, PharmD, MIPH

Academic Editor

PLOS ONE
---

## [Editor Report · Acceptance letter]

14 Jan 2020

PONE-D-19-28324R1 

Dispensing of antibiotics without prescription and associated factors in drug retail outlets of Eritrea: a simulated client method 

Dear Dr. Weldemariam:

I am pleased to inform you that your manuscript has been deemed suitable for publication in PLOS ONE. Congratulations! Your manuscript is now with our production department. 

With kind regards,

on behalf of

Dr. John Rovers 

Academic Editor

PLOS ONE